# The Anabolic Response to Dietary Protein Is Not Limited by the Maximal Stimulation of Protein Synthesis in Healthy Older Adults: A Randomized Crossover Trial

**DOI:** 10.3390/nu12113276

**Published:** 2020-10-26

**Authors:** Sanghee Park, Jiwoong Jang, Myung Dong Choi, Yun-A Shin, Scott Schutzler, Gohar Azhar, Arny A. Ferrando, Robert R. Wolfe, Il-Young Kim

**Affiliations:** 1Lee Gil Ya Cancer and Diabetes Institute, Gachon University, Incheon 21999, Korea; sangheepark1@gachon.ar.kr (S.P.); korea81@hanmail.net (J.J.); 2Department of Molecular Medicine, College of Medicine, Gachon University, Incheon 21999, Korea; 3Gil Medical Center, Gachon University, Incheon 21565, Korea; 4Department of Human Movement Science, Oakland University, Rochester, MI 48309, USA; choi@oakland.edu; 5Department of Prescription & Rehabilitation of Exercise, Dankook University, Cheonan 31116, Korea; shinagel@empal.com; 6Center for Translational Research in Aging and Longevity, Department of Geriatrics, Donald W. Reynolds Institute on Aging, University of Arkansas for Medical Sciences, Little Rock, AR 72205, USA; SESchutzler@uams.edu (S.S.); AzharGohar@uams.edu (G.A.); AFerrando@uams.edu (A.A.F.); RWolfe2@uams.edu (R.R.W.)

**Keywords:** aging, anabolic response, protein synthesis, protein breakdown, stable isotope tracers, essential amino acids

## Abstract

We have recently demonstrated in young adults that an anabolic response with mixed meal protein intake above ~35 g/meal, previously recognized as an “optimal” protein dose, was further stimulated. However, it is unknown if this applies to older adults. We therefore examined anabolic response to a mixed meal containing either 35 g (MOD, moderate amount of protein) or 70 g (HIGH, high amount of protein) in a randomized cross-over metabolic study in older adults (*n* = 8). Primed continuous infusions of L-[^2^H_5_] phenylalanine and L-[^2^H_2_]tyrosine were performed to determine whole-body protein kinetics and muscle protein fractional synthesis rate (MPS) in basal fasted and fed states. Whole-body protein kinetics (NB, net protein balance; PS, protein synthesis; PB, protein breakdown) and MPS was expressed as changes from the baseline post-absorptive state. Consistent with our previous findings in young adults, both feedings resulted in a positive NB, with HIGH being more positive than MOD. Furthermore, NB (expressed as g protein∙240 min) increased linearly with an increasing amount of protein intake, expressed relative to lean body mass. The positive NB was achieved due mainly to the suppression of PB in both MOD and to a greater extent HIGH, while PS was only increased in HIGH. Consistent with the whole-body data, MPS was significantly higher in HIGH than MOD. Plasma concentrations of essential amino acids and insulin were greater in HIGH vs. MOD. We conclude that in the context of mixed meals, whole-body anabolic response linearly increases with increasing protein intake primarily through the suppression of PB, and MPS was further stimulated with protein intake above the previously considered “optimal” protein dose in older adults.

## 1. Introduction

Sarcopenia, a major factor of the fragility syndrome, is defined as progressive decrease of muscle mass, strength, and function. It is considered a strong predictor of disability and mortality in older adults [1,2]. Slowing or preventing the progression of sarcopenia is of upmost importance for maintaining or improving the quality of life for older adults while it may not be possible to completely reverse the progress of sarcopenia. It is well established that intake of dietary protein stimulates an anabolic response [3,4] mainly via the stimulation of protein synthesis by essential amino acids (EAAs). Especially, a moderate to large amount of protein or EAAs intake similarly increases muscle protein synthesis in young and older adults [5,6,7]. The anabolic response in older individuals has been reported to be maximized with consumption of 0.40 g protein/kg BW/meal or ~ 32 g protein/meal for an 80-kg person [8]. However, this conclusion seems to be incomplete for several reasons. First, it was based entirely on muscle protein fractional synthesis rate (MPS) data despite the fact that the whole-body anabolic response is determined by the balance between protein synthesis and breakdown [9,10]. Second, the anabolic response which led to the conclusion was determined with a protein or AAs supplement rather than in the context of a mixed meal, which is more representative of the manner in which dietary protein is normally consumed [11,12]. Last and importantly, when quantifying an anabolic response it is necessary to take into account the entire body protein pool, because more than half of whole-body anabolic response occurs at organs such as gut [9]. With these points in mind, we previously found in healthy young adults that increasing amounts of protein intake induced greater anabolic responses in the context of a mixed meal [13]. Further, the total anabolic response was due not only to a stimulation of protein synthesis (PS), but also to a suppression of protein breakdown (PB) [13]. Consistent with previous dose-response studies [8,14], with respect to the MPS response to dietary protein, we found that MPS was not different following consumption of 40 g or 70 g protein in a mixed meal in young adults [13]. In the present study, we tested hypotheses that, similar to young adults in the our previous study [13], (1) the whole-body net anabolic response would be greater with 70 g than 35 g of protein consumed in the context of a mixed meal; (2) the anabolic response would increase linearly with increasing amounts of dietary protein intake; and (3) MPS would not be different between 35 g and 70 g of protein consumed in a mixed meal in older individuals.

## 2. Materials and Methods

### 2.1. Subjects

Eight healthy older individuals (>60 years) were recruited by using flyers posted around the Little Rock area and the University of Arkansas for Medical Sciences (UAMS) campus (September 2015 through January 2016). Subject eligibility for the study was accessed based on a battery of medical tests including liver and renal function, plasma electrolytes, blood glucose concentration, and medical history tests. Exclusion criteria precluded participants with active malignancy within the past 6 months, a chronic inflammatory disease, diabetes, low hematocrit or hemoglobin concentration, gastrointestinal bypass surgery, low platelets, concomitant use of corticosteroids, and any unstable medical conditions. In addition, participants who performed any types of strenuous physical activity more than once a week were excluded. Prior to study initiation, the ethics committee of the Institutional Review Board at the UAMS approved this study (IRB# 204291) which was performed in accordance with the Declaration of Helsinki and written informed consent was obtained from all subjects. Clinical Trial Registry number and website: NCT03765710, https://register.clinicaltrials.gov/prs/app/action/SelectProtocol?sid=S0008HGW&selectaction=Edit&uid=U0003YFM&ts=2&cx=9zc687.

### 2.2. Experimental Design

Body composition was determined by dual-energy X-ray absorptiometry (QDR4500A; Holologic, Waltham, MA, USA) (Table 1) at the time of study screening for eligibility. In a randomized cross-over design, eligible subjects consumed two different amounts of protein intake in isocaloric mixed meals in a random order with at least 1 week apart between trials: 35 g (MOD) vs. 70 g (HIGH) of protein. As in the previous study [13], beef patties were the main source of protein in the mixed meal. Participants were instructed not to perform strenuous physical activity for >72 h before each metabolic experiment. Meals were provided in the 2-day run-in period for dietary normalization followed by the metabolic experiment on day 3. In the Metabolic Kitchen at the Reynolds Institute on Aging (RIOA), a study dietician prepared all foods (Table 2). Subjects were provided a dietary record and point and shoot digital camera when they received the 2-day meal allotments at RIOA and consumed them in their own convenient places including home. Subjects were instructed to record percentage of the meal consumed and to photograph the meal before and after the consumption. Participants were instructed to return camera on the morning of the third day when they reported to the RIOA for the metabolic experiment following >10 h fasting. The order of the two feeding experiments was randomized using drawing lots.

### 2.3. Preparation of Interventional Meals

Interventional meals were prepared as previously described [13]. Briefly, precooked 85% lean ground beef was purchased from a local grocery to form into patties weighing 128.7 g or 321.7 g. A gas burning stove was used to fully cook the beef patties in a skillet. We individually packaged the cooked beef patties and stored the patties at −18 °C. The non-beef components of the mixed meals which are components of routine such as canned peaches, corn kernel, and rice krispies were prepared in the Metabolic Kitchen in advance. Under refrigeration at 4 °C, the meals were thawed overnight and microwaved immediately before being provided. The protein content was ~19 g per 100 g raw beef. Amino acid composition in the intervention meal of MOD and HIGH was tryptophan (0.21, 0.4), threonine (1.25, 2.58), isoleucine (1.42, 2.95), leucine (2.71, 5.46), lysine (2.45, 5.35), methionine (0.83, 1.72), phenylalanine (1.35, 2.72), valine (1.68, 3.4), and histidine (0.99, 2.13), respectively, in the unit of gram. MOD and HIGH contained total 12.9 g and 26.7 g EAAs, respectively.

### 2.4. Stable Isotope Tracer Infusion Protocol

The 8-h tracer infusion protocol is shown in Figure 1. On the third day, participants reported to the RIOA following an overnight fasting (after 22:00). Before the initiation of tracer infusion, two catheters were inserted into veins of each lower arm; one for the infusion of tracers and the other for “arterialized” blood sampling through a heating box. First blood sample was obtained to determine baseline isotopic enrichments, followed by initiation of a primed-continuous infusion of tracers: L-[^2^H_5_]phenylalanine (prime, 4.60 µmol·kg^−1^; rate, 3.92 µmol·kg^−1^·h^−1^) and L-[^2^H_2_]tyrosine (prime, 0.95 µmol·kg^−1^; rate, 1.57 µmol·kg^−1^·h^−1^) with a priming of L-[^2^H_4_]tyrosine (0.33 µmol·kg^−1^). All isotope tracers were purchased from Cambridge Isotope Laboratories (Andover, MA). To measure tracer enrichment and plasma insulin concentrations and AAs, blood samples were collected throughout the metabolic study at 0, 120, 180, and 210 min prior to a meal consumption (the fasted states) and at 270, 300, 330, 360, 390, 420, 450, and 480 min (the fed states). Muscle samples from vastus lateralis muscles were obtained before meal intake (at 120 and 240 min) and at the termination of the metabolic study (at 480 min) (Figure 1) to determine MPS (%/h) in the fasted and fed states.

### 2.5. Calculations of Protein Kinetics

Whole-body protein kinetics were analyzed according to a 2 pool model, as previously described [15]. Briefly, in the fasted state, appearance rate of phenylalanine, *R_a_* Phe, reflects rate of protein breakdown while disappearance rate of phenylalanine, *R_d_* Phe, reflects the sum of protein synthesis rate and rate of phenylalanine hydroxylation to tyrosine (HYDROX). In the fed state, however, *R_a_* Phe reflects both rate of protein breakdown and rate of appearance of protein from the meal, latter of which was to be subtracted to determine rate of protein breakdown. To convert kinetics at the level of amino acid to the level of protein, kinetic values of phenylalanine were divided by 0.04 with the assumption that Phe contributes 4% to protein in muscle. To determine hydroxylation rate of Phe to tyrosine (Tyr), *R_a_* Tyr was also determined as the product of *R_a_* Tyr and % of *R_a_* Tyr derived from Phe (%*R_a_* Tyr) divided by 100. The specific equations for calculations of whole-body protein kinetics [15,16,17] are listed below:Total appearance rate into plasma (*R_a_*) (µmol/kg/min) = F/EFractional *R_a_* of Tyr from Phe (%/100) = E_Tyr_ M + 4/E_Phe_ M + 5Rate of Phe hydroxylation (HYDROX) (µmol/kg/min) = fractional *R_a_* of Tyr from Phe × *R_a_* Tyr_MPE_Rate of protein synthesis (µmol/kg/min) = [(*R_a_* Phe − HYDROX)/0.04]*R_a_* Exo = (Protein intake × GITD) × (1 fraction of *R_a_* Phe HYDROX)Protein breakdown rate (µmol/kg/min) = [(*R_a_* Phe − F_Phe_)/0.04 − *R_a_* Exo]Net protein balance (µmol/kg/min) = Protein synthesis rate − Protein breakdown rateMPS (%/h) = [(E_BP2_ − E_BP1_)/(E_IC_ × *t*)] × 60 × 100

Enrichment is expressed as tracer to tracee ratio (TTR) (Figure 2) or mole percent excess (MPE), calculated as TTR/(TTR + 1). TTR was used for PB calculation whereas MPE was used for PS calculation. E is enrichment of respective tracers. F is the tracer infusion rate into a venous site: F_Phe_ for phenylalanine tracer. E_Tyr M + 4_ and E_Phe M + 5_ are plasma enrichments of tyrosine and phenylalanine at M + 4 and M + 5 relative to M + 0, respectively. R_a_ Exo is the rate of exogenous amino acids appearing in the circulation as a result of the protein digestion, accounting for gastrointestinal tract digestibility (GITD) according to previously reported values [18,19,20] and the fraction of absorbed amino acids directly hydroxylated before reaching the peripheral circulation. HYDROX is the appearance rate of tyrosine derived from phenylalanine via process of hydroxylation. As previously described [15], MPS was calculated. For the MPS analysis, five out of total eight older adult subjects were included due to issues regarding muscle samples. E_BP1_ and E_BP2_ are the enrichments of protein bound L-[ring-^2^H_5_]phenylalanine from muscle samples at t_1_ and t_2_, respectively, and E*_IC_* is enrichment of intracellular free L-[ring-^2^H_5_]phenylalanine (120, 240, and 480 min). *t* is the duration in minutes elapsed between two muscle biopsies; 60 and 100 are factors used to express MPS in percent per hour.

### 2.6. Analytic Methods

Determination of plasma tracer enrichments was performed by gas chromatography-mass spectrometry (GCMS: Models 7890A/5975; Agilent Technologies, Santa Clara, CA, USA). As previously described [15], muscle tissue samples were prepared by homogenizing with 0.5 mL of 10% sulfosalicylic acid and centrifuged for collecting supernatant. Muscle intracellular free AAs were extracted from 300 µl supernatant fluid by cation-exchange resin columns and dried under Speed Vac. The remaining pellet was hydrolyzed in 3 mL of 6 N HCl at 105 °C for one day. Phenylalanine tracer enrichment from intracellular free and bound tracer in muscle was analyzed as in plasma analyses. Concentrations of plasma amino acid (AAs) were analyzed by liquid chromatography-mass spectrometry (QTrap 5500 MS; AB Sciex, Foster City, CA, USA) as previously described [21]. Concentrations of plasma insulin were analyzed using commercially available human insulin ELISA kit (Alpco Diagnostics, Salem, MA, USA).

### 2.7. Statistical Analysis

A two-tailed student’s t-test was used to compare differences in protein kinetics (NB, PS, and PB), MPS, and area under the curve (AUC) of plasma insulin. All variances we analyzed were homogeneous, which we tested using Levene’s homogeneity of variance test (for all; *p* > 0.15). The analyses of the plasma insulin and AAs including concentrations and sampling time, a continuous variable, were determined by using two-way repeated-measures of ANOVA. If there were significant main effects or interactions, a two tailed student’s t-test was performed for specific comparisons. To compare protein intake to NB in older adults, Pearson’s correlation coefficients and linear regression were performed. *p* < 0.05 was considered to be statistically significant. This analysis was performed using SPSS statistical package (version 24.0; IBM Inc, Chicago, IL, USA) or Microsoft Excel (Microsoft Corporation, CA, USA). All data are presented as mean ± standard error of the mean (SEM). A sample size of subjects in a crossover design was estimated to have 80% power based on the power analysis (two-sample equal variance t-test) of NB, PS, and PB to detect differences in means of 1.68 standard deviations or larger which was based on the previous study we completed in young, healthy adults [13]. We assumed the standard deviations of the older population will be similar.

## 3. Results

### 3.1. Protein Kinetics at Whole-Body and Muscle Levels

Our primary focus was the anabolic responses to different amounts of dietary protein intake in a mixed meal. Both feedings significantly increased the anabolic response, as reflected by NB, and to a greater extent in HIGH vs. MOD (Figure 3). The positive NB was due primarily to a significant suppression of PB in both feedings; only HIGH significantly increased PS (for all; *p* < 0.002). In agreement with our previous study in young adults [13], increasing amounts of protein consumption in the context of the mixed meal showed a linear positive relationship with increase in NB in older adults when protein intake is expressed relative to lean body mass (*p* < 0.001) (Figure 4). Changes in MPS from the fasted state were greater in HIGH compared to MOD (*p* = 0.03) (Figure 5).

### 3.2. Plasma Profile

Plasma amino acid concentration responses are shown in Figure 6 and Figure 7. For the EAAs, leucine, and BCAA, there was a main effect for the amount of protein intake and an interaction effect for the amount of protein intake by time (*p* < 0.001). Following a meal intake, EAAs, leucine, and BCAA were greatly elevated in HIGH but not in MOD (*p* < 0.001) while there existed no difference in NEAAs between MOD and HIGH (Figure 6). Insulin area under the curve was significantly increased in HIGH compared with MOD (*p* = 0.02) (Figure 7).

## 4. Discussion

In the present study, we found (1) a positive NB following both feedings, with HIGH greater than MOD (by 101%); (2) a modest but significant increase of PS only in HIGH above basal fasted states; and (3) a greater postprandial MPS above the basal state in HIGH as compared to MOD. In accordance with our previous findings in young adults [13], we found a positive linear relation between protein intake, normalized for lean body mass, and NB in older adults.

In our previous study in young adults, we showed that in the context of mixed meals, intake of dietary protein (~40 g or ~70 g protein/meal) above the previously considered “optimal” protein intake (e.g., ~0.24 g/kg/meal or ~20 g protein/meal for an 80-kg person) further stimulated the whole-body anabolic response, but not MPS [13]. In the current study, we tested if the same responses occurred in older adults. The current study question was derived in part from previous studies demonstrating that maximal anabolic response in muscle (i.e., MPS) can be achieved with consumption of high-quality protein of 0.4 g/kg/meal (e.g., ~32 g protein/meal for an 80-kg person) in older adults [8]. However, this conclusion is limited by several factors, including: (1) it was entirely based upon MPS, whereas the anabolic response is the balance between MPS and MPB (muscle protein breakdown); (2) MPS was determined following consumptions of a high quality pure AA/protein, which is not analogous to mixed meal protein intake which is how most people consume protein; and (3) more than half of protein turnover occurs in tissues other than muscle [9,22], indicating that muscle specific anabolic response does not represent the whole-body anabolic response (i.e., total anabolic response). In accordance with our previous study in young adults [13], we demonstrated that the stimulation of whole-body anabolic response is not limited by the maximal stimulation of MPS, but instead increases linearly with increasing amounts of mixed meal protein intake in older adults. The increased anabolic response following feeding in the present study was largely due to a suppression of PB, which agrees with our previous findings [11,12,13,15]. However, we found that PS was stimulated above basal fasted states with HIGH but not MOD. These results are contrary to our previous findings in young adults, where both feedings resulted in a simulation of PS, but to a greater extent with the higher dose [13]. The current study findings point to an anabolic resistance at the whole-body level in older adults in terms of protein synthesis [6,23,24].

The potential mechanisms responsible for different levels of anabolism following a mixed meal containing varying amounts of protein may involve two main factors: (1) increased peripheral EAA availability and (2) hyperinsulinemia. First, HIGH induces increased EAA availability in the plasma and inward transport into the intracellular compartment [25]. Thus, an increase in PB is not required to maintain intracellular EAAs [26]. Our previous studies [12,13,27] have demonstrated that higher extracellular EAA concentration in HIGH induced higher intracellular AA availability, which plays an important role in further suppression in PB [25]. The greater responses of leucine and total EAAs may explain why only HIGH increased PS at the whole-body level [3,5] (Figure 3). The fact that there was no change in EAAs and leucine following MOD is at odds with our previous studies in young subjects, in which 38 g of protein intake in a mixed meal induced a significant elevation of leucine [15] and EAAs [13]. The discrepancy in responses between young and older adults may be due to different sources of interventional protein and/or age-related changes in splanchnic amino acid retention [5,28].

The increase in insulin concentrations in response to the mixed meal may also have impacted the total anabolic response. Insulin can stimulate an anabolic state by activating PS and/or suppressing PB [29]. The greater insulin response with HIGH is consistent with the kinetic data (i.e., greater suppression of PB and stimulation of PS with HIGH vs. MOD). However, this is contrary to our previous study in young adults, in which the insulin responses following the mixed meal were not different between the protein doses. It is not clear why the insulin response following HIGH in the current study was greater than MOD, despite the smaller carbohydrate content of the mixed meal (Table 2). Although a greater EAA content in HIGH can stimulate insulin secretion, it is unlikely that EAA is more potent than carbohydrate in stimulating insulin secretion for a given calorie content [30]. Thus, the role of the higher insulin response in the current study is also unclear. In the previous study in young adults, the higher protein dose resulted in a greater suppression of PB despite similar insulin responses in the two meals. Furthermore, GreenHaff et al. [31] showed that PB was significantly suppressed with increasing insulin concentrations up to 30 µIU/mL, with no further suppression in PB above this level. In the present study, both feedings induced an insulin response above 30 µIU/mL.

Finally, we found that the amount of protein intake was linearly associated with NB in older adults, when intake was normalized for lean body mass [13]. This observation is consistent with our previous finding in young adults [13]. In addition, we found that protein intake (i.e., HIGH of 70 g vs. MOD of 35 g) above 32 g of protein (or 0.4 g/kg/meal) further stimulated MPS in the context of a mixed meal in older adults. The finding is contrary to the findings from previous studies showing no further stimulation in MPS with consumption of protein intake above 32 g in older adults [8,23] and young adults [13]. While the discrepancy cannot be unequivocally explained, it is likely due to the attenuated plasma EAA responses following consumption of dietary protein in the context of mixed meals compared to young adults or those following consumption of pure protein/AA, and due partly to insulin-mediated suppression of protein breakdown. The attenuated plasma EAA responses may require more dietary protein/AA consumption to induce a comparable hyperaminoacidemia for stimulation of MPS.

Our study has limitations. First, we have examined the acute anabolic response, which does not guarantee meaningful changes in muscle mass and/or strength over time. However, it is important to appreciate that long-term changes in strength and muscle mass are the cumulative result of acute anabolic responses. Second, the sample size of the current study was relatively small (*n* = 8), although the sample size was determined based on power analysis, and was sufficient to find statistically significant results. A longer-term study with more subjects is warranted to address these limitations empirically.

## 5. Conclusions

Our principal conclusion is that in older individuals a level of protein intake of 70 g in the context of a mixed meal induces a greater anabolic response than when an isocaloric meal contains 35 g protein. The higher protein intake stimulated a greater protein synthetic response, both in the whole-body and skeletal muscle. Whole-body PB was also suppressed to a greater extent with the higher level of protein intake. These data indicate that the previously proposed “optimal” level of dietary protein in a meal of 30–35 g significantly underestimates the true value for older individuals.

## Figures and Tables

**Figure 1 nutrients-12-03276-f001:**
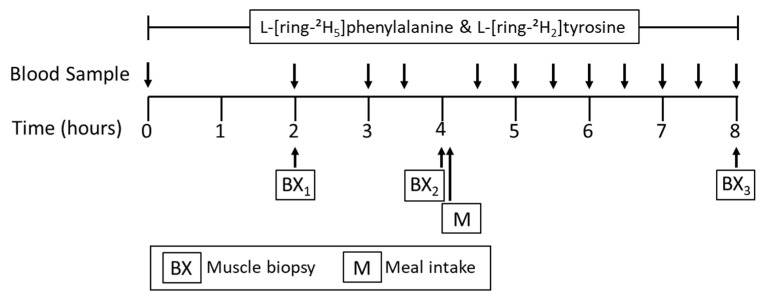
Tracer infusion protocol for metabolic study.

**Figure 2 nutrients-12-03276-f002:**
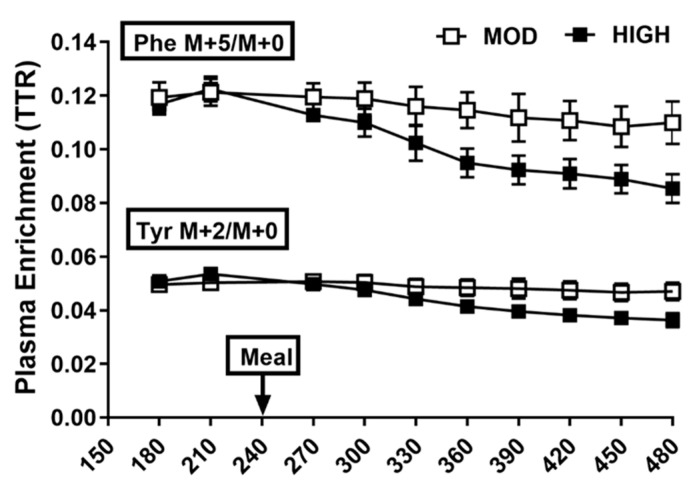
Plasma enrichments of tracers of phenylalanine (Phe M5) and tyrosine (Tyr M2) before and after the meal consumption containing 35 g (MOD) or 70 g (HIGH) of protein. Values are expressed as means ± SEM. TTR, tracer to tracee ratio (*n* = 8).

**Figure 3 nutrients-12-03276-f003:**
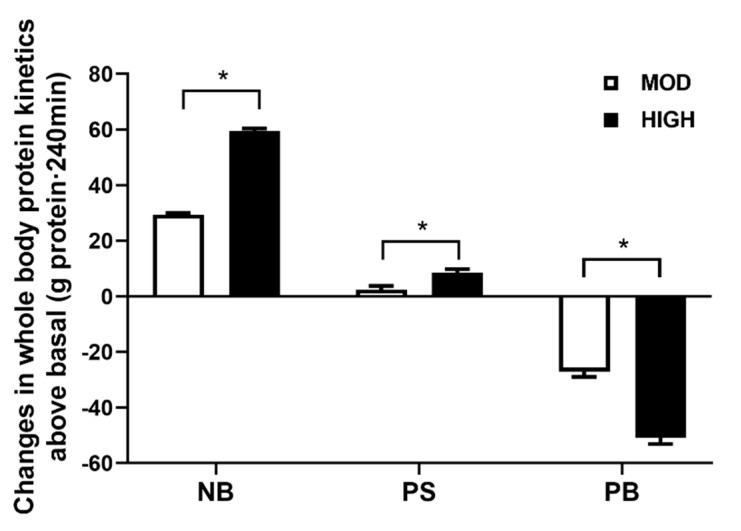
Changes in rates of whole-body protein net balance (NB), synthesis (PS), and breakdown (PB) above basal fasted states following consumption of a meal containing either 35 g (MOD) or 70 g (HIGH) of dietary protein. * Significantly different from MOD, *p* < 0.002. Values are expressed as means ± SEM (*n* = 8).

**Figure 4 nutrients-12-03276-f004:**
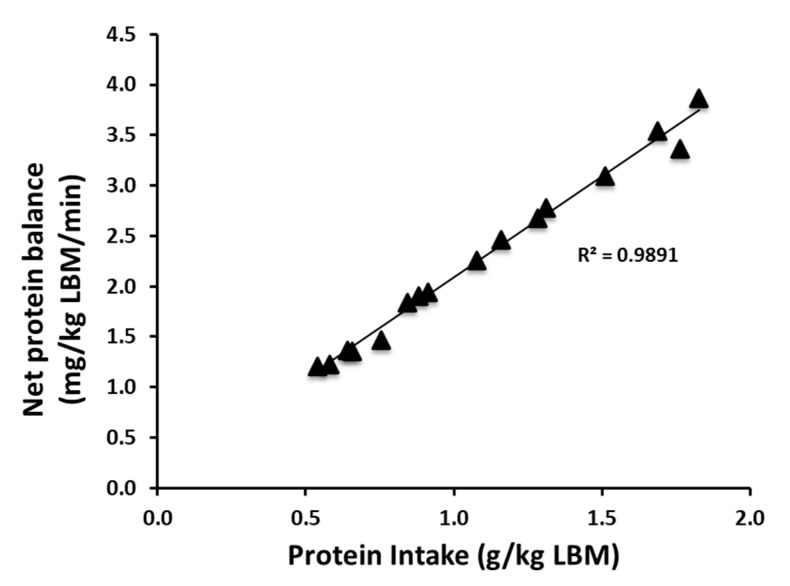
The correlation between the increasing amounts of protein intake and whole-body net protein balance following a meal intake. The correlation was statistically significant, *p* < 0.001 (*n* = 8).

**Figure 5 nutrients-12-03276-f005:**
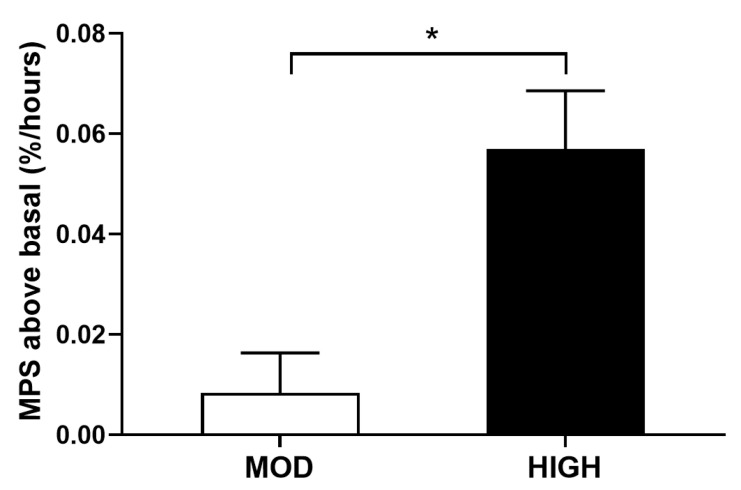
Muscle protein fractional synthesis rate (MPS, %/hours) above fasted states following a meal consumption containing either 35 g (MOD) or 70 g (HIGH) of dietary protein. * Significantly different from MOD, *p* = 0.03. Five subjects were included for MPS analysis due to muscle sample issues. Values are expressed as means ± SEM (*n* = 5).

**Figure 6 nutrients-12-03276-f006:**
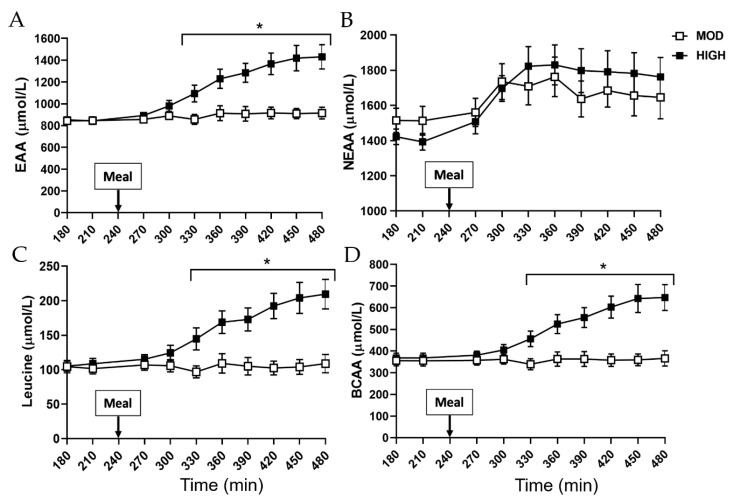
Plasma concentrations of (**A**) total essential amino acids (EAAs), (**B**) nonessential amino acids (NEAA), (**C**) leucine, and (**D**) branched chain amino acids (BCAAs) before and after consumption of a meal containing either 35 g (MOD) or 70 g (HIGH) of dietary protein. * Significantly different from MOD, *p* < 0.001. Values are expressed as means ± SEM (*n* = 8).

**Figure 7 nutrients-12-03276-f007:**
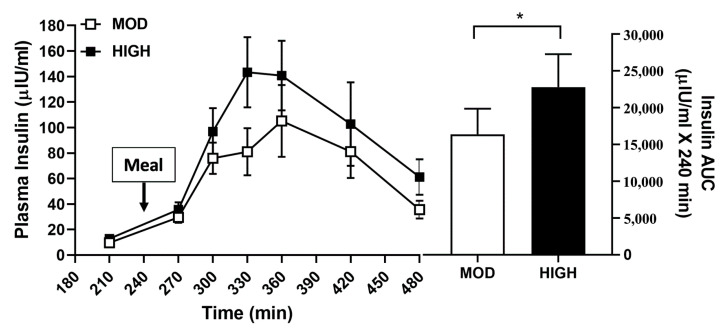
Plasma concentrations of insulin before and after consumption of a meal containing 35 g (MOD) or 70 g (HIGH) of dietary protein. * Significantly different from MOD, *p* = 0.02. Values are expressed as means ± SEM (*n* = 8).

**Table 1 nutrients-12-03276-t001:** Subject characteristics.

Subjects (M/F)	8 (4/4)
Age, year	69.3 ± 1.8
Weight, kg	82.9 ± 4.9
BMI, kg/m^2^	27.4 ± 0.9
LBM, kg	49.9 ± 3.5
Fat mass, %	34.5 ± 2.3

BMI, body mass index; LBM, lean body mass; M/F, the No. of male and female subjects. Values are expressed as means ± SEM.

**Table 2 nutrients-12-03276-t002:** Macronutrients of 2-day run-in meal on day 1–2 and metabolic study on day 3.

		Run-in Foods on Day 1–2								
Protein Levels	Energy Intake, Kcal	Protein			Fat				CHO	
		g	%		g		%		g	%
MOD	2324 ± 135	83.1 ± 4.9	14.1 ± 0.3		91.0 ± 5.6		34.7 ± 0.3		301.0 ± 16.7	51.2 ± 0.2
HIGH	2328 ± 135	83.2 ± 4.8	14.1 ± 0.3		91.7 ± 5.6		34.9 ± 0.2		300.5 ± 16.9	51.0 ± 0.2
**Interventional Meals of Metabolic Infusion Study on Day 3**										
**Protein Levels**	**Meal, Kcal**	**Beef Protein, g**	**Protein**		**Fat**		**CHO**		**Nonprotein Energy, %**	
			**g**	**%**	**g**	**%**	**g**	**%**	**Fat**	**CHO**
MOD	1100	23.6	35.7	12.9	41.3	33.6	147.9	53.5	38.6	61.4
HIGH	1100	59	70.3	25.9	36.2	30	119.6	44.1	40.5	59.5

MOD, moderate amount of protein; HIGH, high amount of protein; CHO, carbohydrate. Values are expressed as means ± SEM (*n* = 8).

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
