# Peer review of "The Anabolic Response to Dietary Protein Is Not Limited by the Maximal Stimulation of Protein Synthesis in Healthy Older Adults: A Randomized Crossover Trial"

_nutrients, 2020, doi:10.3390/nu12113276_

Round 1

Reviewer 1 Report

STRUCTURE: The manuscript is properly structured.

TITLE: The title should inform about the type of study.

INTRODUCTION:

  • The relationship between sarcopenia and protein intake should be better explained.
  • Line 52: Add reference.
  • Line 57-60-62: It is recommended not to use the first person
  • Why you have this hypothesis “the whole-body net anabolic response would be greater with 70 g than 35 g of protein consumed in the context of a mixed meal” if you previously mentioned that we found that MPS was not different following consumption of 40 g or 70 g?

MATERIAL AND METHODS:

  • Line 80-81: The link to the Clinical Trial Registry must be properly placed or removed
  • Table 1. What is SE? All abbreviations should be explained.
  • Line 88 “As in the previous study” -- Which study is referred to? Add reference.
  • The description of trial design (such as parallel, factorial) including allocation ratio, is missing.
  • How sample size was determined
  • How the sample was distributed?
  • It is recommended not to attach the tables as photos. The quality will be lost.
  • Who performed the extractions? The same people who carried out the research? People outside of the research? Specify.
  • Reference the calculations and equations used where they come from.
  • How long did the recruitment and intervention take?
  • How was the homogeneity of the sample calculated?

DISCUSSION:

  • Authors should compare with other studies, not just their own with different samples.
  • Why were the same protein sources not used in both studies so that it would make more sense to compare the results?
  • Explain possible causes of different outcomes in Young adults vs. older adults.
  • The study has no limitations? Trial limitations, addressing sources of potential bias, imprecision, and, if relevant, multiplicity of analyses are needed.
  • Generalizability (external validity, applicability) of the trial findings

Author Response

Dear Reviewer,

Thank you for taking your valuable time to review our manuscript and your comments.

Sincerely,

Reviewer 2 Report

Nutrients-961877-Peer Review-V1 Oct 6 2020

Title: The anabolic response to dietary protein is not limited by the maximal stimulation of protein synthesis in healthy older adults

Authors: S Park, J Jang, MD Choi, YA Shin, S Schutzler, G Azhar, AA Ferrando, RR Wolfe, IlY Kim

The researchers examined whether anabolic response to a mixed meal containing either 35g (MOD) or 70g (HIGH) in a cross-over metabolic study in 8 older adults (age >60 years). Whole-body protein kinetics and muscle protein fractional synthesis rate (MPS) were assessed in basal fasted and fed states. Whole-body protein kinetics and MPS were expressed as change from the baseline post-absorptive state. In consistence with the researcher’s earlier findings in young adults, both feedings resulted in a positive NB (Net Protein Balance), with HIGH being more positive than MOD. NB increased linearly with an increasing amount of protein intake, expressed relative to lean body mass. The positive NB was achieved due mainly to the suppression of PB (Protein breakdown) in both MOD and HIGH groups, while PS (Protein Synthesis) increased only in HIGH group. Consistent with the whole-body data, MPS was significantly higher in HIGH than MOD. Plasma essential amino acids and insulin concentrations were greater in HIGH vs. MOD. The researchers concluded that in the context of mixed meals, whole-body anabolic response linearly increases with increasing protein intake primarily through the suppression of PB, and MPS was further stimulated with protein intake above the previously considered “optimal” protein dose in older adults.

The study is indeed remarkably interesting; however, the “n” number is exceptionally low. The study was done only in 8 older adults (>60 years), so a power analysis will be extremely difficult.

Please include the statistical power analysis data in the manuscript.

Include the digestive health of the subjects after taking 70g (HIGH) diet.

Include the adverse events evaluated in this investigation.

The manuscript may be considered after satisfactory revision.

Author Response

(The authors gave the same response as above.)

Reviewer 3 Report

The article “The anabolic response to dietary protein is not limited by the maximal stimulation of protein synthesis in healthy older adults”, is an interesting and generally well-written article.  Having said this, I have some concerns:

Major concerns:

-the article appears to be the summary of 8 patients, however there is no mention of sample size determination or power analysis, that should be included in the statistical analysis

-the abstract is not formatted in the conventional way, listing background, objective, methods (including study design, i.e. repeated measures longitudinal, sample size, site, population), results, discussion and conclusion & rarely (never), would you list previous work, where the studies with young adults is noted x2, unconventional, this should be amended to focus on the current results only, previous work can be integrated into the discussion.

-there are many self-citations, almost half, is this appropriate and needed?

-the discussion lies heavily on the topic and relationship to insulin, was this measured, and if so where was it reported.

-all tables and figures should have the N= listed clearly for the readers, as this is not clear.

t-he article appears to be a copy methodologically to that of ref#9 Kim et.al. 2016, just with a different patient population, which in and of itself may not be an issue, however puts into question the innovation of the work.

Minor concerns:

-how did researchers determine from 40 g in previous work to 35 g in this study…

-Table 2:  all acronyms should be listed in the table description, i.e. MOD, HIGH, CHO

-the methods are a bit confusing, line 93-94 they ate at the RIOA, or it was delivered at home?, also line 102-103, what were the “non-beef components” and is this important to know?  Perhaps clarify.  Were patients financially compensated?

-Figures 6, 7.  For clarity the sub-images should be annotated, A, B, ect…and accordingly footnoted in the figure legend/description.  Figure 6, all x-axis are missing their description including units.  Figure 7. Appears as one figures, should be broken-up into A, and B.

Specifically:

-Little Rock is listed both in line 68 and 70, this repetitious

-Line 110 (after 22:00) semi-colon missing.

-Line 112 was, should read were

-section 2.5 of the methods, this may be out of scope for the average reader, perhaps could provide additional context.

-the use of single word acronyms is unusual, “E” and “F” line 149, small caps line 160 “t”…

Author Response

(The authors gave the same response as above.)

Reviewer 4 Report

  1. Please introduce the abbreviations at their first appearance in the main text.
  2. What do you consider as "regular exercise"? Are long walks included in this or cycling?
  3. Please clearly annote the number of samples in each group for each analysis. Were it samples from 3 individuals for each? After all, the sample size is extremly low which rises questions about validity and reproducibility of results.
  4. What is the AA profile of the tested diet. Was it tested?
  5. Proposed research is tendentious. According to disclousure, one of the authors have previously received grants and honoraria from cattle industry. Research does not take into account the overall health-related aspects of nutrition and what the right source of protein is. Beef is neither the best source of protein from purely nutritious point of view, nor is it in any way environmentaly acceptable.
  6. The article fails to mention ANY limitation of the current study, though clearly it has several.
  7. Finally, participants are overweight with very low (?) overall physical activity (apparently it had not been quantified). Rising the amount of protein in their diet does not seem to be the solution of the problem. The real improvement requires a more hollistic approach which takes into account physical activity, overall diet (quality, right amount of ALL nutrients - not only proteins). Moreover, health risks related to high protein intake should be considered.

Author Response

(The authors gave the same response as above.)

Round 2

Reviewer 1 Report

No further comments. 

Author Response

Dear Reviewer,

We appreciate your time and effort to review our manuscript.

Sincerely,

Sanghee Park

Reviewer 2 Report

The revised manuscript is recommended for acceptance

Author Response

(The authors gave the same response as above.)

Reviewer 3 Report

None.

Author Response

(The authors gave the same response as above.)

Reviewer 4 Report

  1. At least 18 of 31 citations are self-citations. This issue was rised by other reviewers, but it was not addressed appropriately by authors. Authors seem to consider this normal and acceptable - it is so much easier to discuss with yourself instead of others. In fact, the authors seem to address the reviewers' comments in similar way - by picking what was easy while nearly ommitting all problematic issues.
  2. It is still not clear how many samples were used in each subgroup for each Analysis. I would strongly suggest providing the N value not for whole Analysis but for each Group on the graph. There should be no doubts as to the number of samples analysed.
  3. "Our study has potential limitations" - those are not potential, but real limitations. Still, some of the limitations were clearly not listed. Studied Group was not purely older adults, those were obese or overweight older adults with sedentary Lifestyle. This is an important difference. From metabolic (and medical) perspective active and slim 80 years old subject is "younger" than overweight 80 years old subject who prefers sedentary lifestyle.

Author Response

Thank you for your time to review our manuscript and comments.

Comment 1: At least 18 of 31 citations are self-citations. This issue was rised by other reviewers, but it was not addressed appropriately by authors. Authors seem to consider this normal and acceptable - it is so much easier to discuss with yourself instead of others. In fact, the authors seem to address the reviewers' comments in similar way - by picking what was easy while nearly ommitting all problematic issues.

Response 1: We believe that it is fine as the other reviewers were satisfied with our response. Again, our laboratory has been involved in this type of research for more than 40 years, so it is not surprising that much of the previous work on the topic has been done in our laboratory. Indeed, one of the main goals of this paper was to compare the responses in older individuals with our previously- published results in younger subjects. Extensive quoting of our own work was therefore mandatory. We are unaware of relevant papers to which we have “omitted reference”, but if the reviewer is aware of such papers we will be happy to discuss our findings in relation to those findings.

Comment 2: It is still not clear how many samples were used in each subgroup for each Analysis. I would strongly suggest providing the N value not for whole Analysis but for each Group on the graph. There should be no doubts as to the number of samples analysed.

Response 2: We added sample size (n) in each figure and table. As a reminder, this study was a crossover design in which each subject completed entire trials (i.e., 2 experimental trials) and all 8 subjects had actually completed 2 trials (MOD and HIGH) in the current study. In each figure or table, if n = X is shown, this means that X number of subjects were included in the analysis (thus, in the respective figure or table).

Comment: "Our study has potential limitations" - those are not potential, but real limitations. Still, some of the limitations were clearly not listed. Studied Group was not purely older adults, those were obese or overweight older adults with sedentary Lifestyle. This is an important difference. From metabolic (and medical) perspective active and slim 80 years old subject is "younger" than overweight 80 years old subject who prefers sedentary lifestyle.

Response 3: We have dropped the word “potential" with regard to limitations (line 314)